# Identification of Crustacean Female Sex Hormone Receptor Involved in Sexual Differentiation of a Hermaphroditic Shrimp

**DOI:** 10.3390/biom13101456

**Published:** 2023-09-27

**Authors:** Fang Liu, An Liu, Haihui Ye

**Affiliations:** Fisheries College, Jimei University, Xiamen 361021, China; liufang@jmu.edu.cn (F.L.); liuan@jmu.edu.cn (A.L.)

**Keywords:** crustacean female sex hormone, interleukin-17 receptor, protandric simultaneous hermaphroditism, crustaceans, sexual differentiation

## Abstract

The neurohormone crustacean female sex hormone (CFSH) contains a highly conserved interleukin-17 (IL-17) domain in the mature peptide. Although CFSH has been demonstrated to stimulate female sexual differentiation in crustaceans, its receptors (CFSHR) have been poorly reported. The present study identified an IL-17 receptor (named *Lvit-*IL-17R), a candidate of CFSHR, from the protandric simultaneous hermaphroditic (PSH) shrimp *Lysmata vittata* through GST pulldown assays and RNAi experiments. *Lvit*-IL-17R is a transmembrane protein with an SEFIR (similar expression as the fibroblast growth factor and IL-17R) domain, as determined through sequence analysis. A GST pulldown experiment confirmed the interactions between the type I CFSHs (CFSH1a and CFSH1b) and *Lvit*-IL-17R. Meanwhile, the RNAi results revealed that *Lvit*-IL-17R displays similar functions to type I CFSHs in regulating sexual differentiation and gonad development. In brief, *Lvit*-IL-17R is a potential receptor for type I CFSHs aimed at regulating the sexual differentiation of the PSH species. This study helps shed new light on the mechanism of sexual differentiation among crustaceans.

## 1. Introduction

The sexual systems in decapod crustaceans are diverse [1,2]. Most species exist as functional females or males throughout their lives [1]. In rare cases, some crustaceans may transform into an intersex form under the effect of parasitic infections, abnormal environmental sex determination, genetic abnormity, and pollution [3]. Notably, some or all Caridean species are naturally intersex [1]. These intersex species first mature as males with ovotestis containing both testicular and ovarian morphologies [2,4,5] and can be classified into two types.

In species that undergo complete sex reversal (e.g., strictly sequential protandric hermaphroditism, or SPH), female germ cells are located on the inner side of the ovotestis and are surrounded by male germ cells. As the female germ cells mature, testicular morphologies completely degenerate, and the ovotestis transforms into an ovary [4,5]. In species that eventually possess both male and female reproductive functions (e.g., protandric simultaneous hermaphroditism or PSH), a relatively obvious boundary divides ovotestis into the ovarian region on the anterior sides and the testicular region on the posterior sides (in the dorsal view). Additionally, the testicular region partially degrades as the ovarian regions mature [2,6,7].

The insulin-like androgenic gland hormone (IAG) and crustacean female sex hormone (CFSH) regulate sexual differentiation in both dioecious and hermaphroditic species [8,9,10,11]. IAG, also known as androgenic gland hormone (AGH), is an insulin-like peptide that was first identified in the androgenic gland (AG) of woodlice (*Armadillidium vulgare)* [12,13]. In decapod crustaceans, IAG was first reported by Manor et al. in 2007 [14]. As silencing IAG expression could ubiquitously block spermatogenesis and/or the development of male secondary sexual phenotypes in decapod crustaceans, IAG is also known as the sexual “IAG-switch” [9,11,15,16,17]. Recent studies have reported that IAG regulates male sexual differentiation via IAG receptors, whose structures resemble insulin-like receptors [18,19,20,21].

On the other hand, CFSH is a neurohormone that was first identified in Atlantic blue crabs (*Callinectes sapidus)* [22]. This hormone regulates the development of female secondary sexual phenotypes, as demonstrated in Atlantic blue crabs (*C. sapidus)*, Chinese mitten crabs (*Eriocheir sinensis)* and peppermint shrimps (*Lysmata vittata)* [10,22,23]. With the rapid advancement in sequencing technology, CFSH transcripts have been identified in various species [24,25,26,27,28,29,30]. Based on the amino acid sequence of deduced mature peptides, two CFSH homologs have been isolated and characterized [27,30]. The type I CFSH possesses eight conserved Cysteine residues and a single conserved N-glycosylation motif (Asn-Xaa-Ser/Thr) in the N-terminal regions. The type II CFSH contains nine or ten cysteine residues, with no N-glycosylation motif [28,29]. Furthermore, a conserved interleukin-17 (IL-17) domain is present in both CFSHs types [27,30].

IL-17 is a specific group of cytokines that are crucial in the hosts’ defense against microbial organisms and the development of inflammatory diseases widespread throughout the animal kingdom [31,32,33]. After IL-17 was first identified in purple sea urchins, *Strongylocentrotus purpuratus* [34,35], this cytokine and its signaling pathways related to specific molecules have been reported in other invertebrates [31,36]. However, IL-17 was not reported in decapod crustaceans until a recent investigation identified IL-17 homologs in Pacific whiteleg shrimps, *Penaeus vannamei* [31]. When these IL-17 proteins were compared with other identified CFSHs, they were considered as CFSH of the species [30]. Recently, a candidate CFSH receptor was identified in mud crabs, *Scylla paramamosain*, which have a conversed SEFIR (similar expression as the fibroblast growth factor and IL-17Rs) domain [37]. Thus, it is reasonable to consider that CFSH is indeed an IL-17 homolog in decapod crustaceans, and CFSH receptors may also share similar structures with the IL-17R of other species.

The peppermint shrimp *L. vittata* is a small caridean species from the genus *Lysmata*, of which most shrimps are popular ornamental species and display a unique PSH sexual system [2,38]. It is also suggested as a good model organism for PSH shrimps for its impressive reproductive fecundity and short generation time [11]. Previous studies have demonstrated that two type I CFSHs (*Lvit*-CFSH1a and *Lvit*-CFSH1b) act in concert in regulating the development of the female external features of the species [10]. Furthermore, because of a regulatory feedback loop between type I CFSH and IAG, CFSHs also inhibit IAG expression to suppress the male sexual differentiation of the PSH species [10]. In the present study, the cDNA of *Lvit*-IL-17R was cloned to explore expression profiles. A glutathione s-transferase (GST) pulldown assay was employed to detect the interaction between *Lvit-*CFSH1a/CFSH1b and *Lvit-*IL-17R. Short-term and long-term silencing experiments were then performed to clarify the functional relevance between *Lvit-*CFSH1a/CFSH1b and *Lvit-*IL-17R in the sexual differentiation of the species.

## 2. Materials and Methods

### 2.1. Animals

*L. vittata* were bred in captivity at the Fisheries College of Jimei University maintained at 25–27 °C, a salinity of 30–32 PSU, and a 12 h light/dark cycle. *L. vittata* were provided with a commercially formulated shrimp diet on a daily basis. The gonadal development stages were defined according to Chen et al. 2019 [6]. 

### 2.2. cDNA Cloning of Lvit-IL-17R

TRIzol^®^ reagent (Invitrogen, Carlsbad, CA, USA) was utilized to extract the total RNA from various tissues according to the manufacturer’s protocol. An *Lvit-*IL-17R fragment was obtained from a transcript library. The 5′ -untranslated region (5′ UTR) was obtained with the SMART TM RACE cDNA Amplification Kit (Clontech, Palo Alto, CA, USA) according to the manufacturer’s protocol and the rapid amplification of cDNA ends (RACE). Coding sequences (CDS) were verified using a polymerase chain reaction (PCR) with LA-Taq polymerase (TaKaRa, Dalian, China) under standard PCR conditions. The PCR primers are listed in Table A1.

### 2.3. The qRT-PCR Assays

The primers used for quantitative real-time PCR (qRT-PCR) were either designed with the Beacon Designer 8.21 software or sourced from a previous study [11]. RT-PCR products were sequenced to ensure accuracy. The amplification efficiency of the primer pairs was also tested. qRT-PCR assays were performed with TB Green Premix Ex Taq II (2X) (TaKaRa, Dalian, China) according to the manufacturer’s protocol. 

### 2.4. Expression Profiles of Lvit-IL-17R

Reverse transcription-PCR (RT-PCR) and qRT-PCR were performed to explore the spatial expression features of *Lvit-IL-17R*.

First, various tissues at gonadal development stage II were dissected to detect the expression profile of *Lvit-IL-17R* in various tissues. Following total RNA extraction and first-strand cDNA synthesis, RT-PCR was performed with Ex-Taq polymerase (TaKaRa, Dalian, China) under standard PCR conditions. *Lvit-β-actin* (GenBank accession no. MT114194) was used as a positive control. PCR products were imaged and photographed using the Gel Image System (Tanon 2500B).

qRT-PCR was then performed to detect the temporal expression profiles of *Lvit-IL-17R* during gonadal development. The androgenic gland (AG), ovarian regions and the hepatopancreas were collected at different developmental stages (n = 4–5). Total RNA extraction and qRT-PCR assays were performed as described earlier.

### 2.5. GST Pulldown Assays

GST pulldown assays were conducted to detect the interactions between *Lvit-*CFSH1a/CFSH1b and *Lvit-*IL-17R. CFSH1a and CFSH1b recombinant proteins were expressed and purified with 6 × His tag as previously described [10]. The recombinant protein of the *Lvit-*IL-17R extracellular domain (rIL-17R) with 6 × His and GST tags was also purified using the prokaryotic expression system. The *Lvit-IL-17R* fragment was inserted into the PET-GST vector with EcoR I and Nhe I restriction enzyme sites and was transformed into *E. coli* TransB (DE3) for prokaryotic expression. After being induced at 16 °C for 20 h (isopropyl-beta-D-thiogalactopyranoside, IPTG, and 0.5 mM final concentration were added), bacterial cells were harvested. The purification of rIL-17R was then conducted using Glutathione Sepharose 4B (Solarbio, Beijing, China) following the established protocol from the supernatant of crude cell extracts. GST (with 6 × His tag) was expressed and purified using the same method as that used on the negative control after transforming the pET-GST vector into DE3.

After reloading rCFSH into the Glutathione Sepharose 4B (Solarbio, Beijing, China), rCFSH1a/rCFSH1b was added and incubated at 4 °C for 1 h. Next, unbound proteins were removed by washing them with 10 mM phosphate-buffered saline (PBS; pH 7.4). One column volume of elution buffer (50 mM Tris-Cl, 10 mM reduced glutathione, and pH 8.0) was added, and the samples were incubated for 10 min. Supernatant-bound proteins were collected through centrifugation and analyzed using SDS-PAGE and Western blotting with anti-His mouse monoclonal antibody.

### 2.6. Short-Term Silencing Experiment

A knockdown experiment was carried out by injecting double-stranded RNA (dsRNA) into the shrimps. A specific fragment of *Lvit-IL-17R* was then selected and cloned into a pGEMT-Easy vector. Next, dsRNA was synthesized with T7 and SP6 RNA Polymerase according to the manufacturer’s instructions. Furthermore, the dsRNA of *green fluorescent protein* (*GFP*) was synthesized as the negative control.

Synthetic dsRNA was diluted with 10 mM PBS (pH 7.4) prior to injection. Shrimps (carapace length: 3.15 ± 0.17 mm; body weight: 48.65 ± 5.94 mg) at stage I of growth were equally and randomly assigned into three treatment groups (n = 5). They were injected with ds *IL-17R* (2 μg/g), ds *GFP* (2 μg/g), or an equal volume of dilution solution (10 mM PBS, pH 7.4). Twenty-four hours post-injection, the shrimps were anesthetized on ice for 5 min.

Samples from the AG, ovarian regions, and hepatopancreas were then collected to examine knockdown efficiency via dsRNA injection. Expression levels related to male sexual differentiation (*IAG1*, GenBank accession number: MT114196; *IAG2*, GenBank accession no. MT114197), and ovarian development (*vitellogenin* (*Vg*), GenBank accession number: MT113122; *vitellogenin receptor* (*VgR*), GenBank accession number: MT114195) were also detected using qRT-PCR.

### 2.7. Long-Term Silencing Experiment

For the long-term silencing experiment, shrimps (carapace length: 3.01 ± 0.11 mm; body weight: 43.47 ± 4.38 mg) (n = 13) at stage I of growth were treated with ds IL-17R (2 μg/g), ds *GFP* (2 μg/g), or an equal volume of dilution solution (10 mM PBS; pH 7.4). During the 36-day experimental period, the shrimps were subjected to an injection once every 4 days (10 injections in total). Twenty-four hours after the tenth injection, the shrimps were anesthetized on ice, and the carapace length and body weight of each shrimp were recorded.

Before tissue collection, the gonad and external features of the male and female external features were photographed. Samples of the AG, the ovarian region, and the hepatopancreas were collected to examine the knockdown efficiency and the effects of *IL-17R* silencing on the expression levels of genes related to sexual differentiation and ovarian development. The parts of gonad tissues (testicular and partial ovarian regions) were fixed in modified Bouin’s Fixative Solution (Phygene, Fuzhou, China) at 4 °C for 24 h. Following the process of gradient alcohol dehydration and paraffin embedding, tissue blocks were sliced into 6 μm sections for hematoxylin and eosin (H & E) staining.

### 2.8. Bioinformatics and Statistical Analyses

In addition to the qRT-PCR primers, we used Primer 5.0 software to design the other primers in this study. ORF Finder (https://www.ncbi.nlm.nih.gov/orffinder/ (accessed on 5 April 2023)) was employed to predict the open reading frame (ORF). SMART (http://smart.embl-heidelberg.de/ (accessed on 5 April 2023)) was used to predict the signal peptides and transmembrane domains. MEGA7 software was used to generate evolutionary trees.

Statistical analyses were conducted using SPSS 18.0 software. All data display a normal distribution, as determined by the Kolmogorov–Smirnov test. The homogeneity of variances was subsequently assessed using Levene’s test. Statistical significance was assessed through the utilization of one-way ANOVA, followed by Tukey’s multiple range tests, with a significance level set at *p* < 0.05. The *F* values of one-way ANOVA analysis are shown. All data are represented as mean ± SD.

## 3. Results

### 3.1. Sequence Analysis of Lvit-IL-17R

The ORF of *Lvit-IL-17R* is 1,878 bp (GenBank accession number: MZ367742), which encodes a 625-aa precursor peptide covering a predicted 19-aa signal peptide, a 295-aa extracellular segment, a 23-aa transmembrane region, and a 288-aa intracellular segment (Figure A1). A 98-aa SEFIR domain was predicted in the intracellular regions (Figure A1). *Lvit*-IL-17R had 37.9% amino acid similarity with the *Sp*-IL-17R (GenBank accession number: ON787957), with 38.4% amino acid similarity in the SEFIR domain.

A phylogenetic tree was constructed with *Lvit-*IL-17R, the SEFIR domain-contained proteins of crustaceans, and IL-17Rs from various categories (Figure 1A). Phylogenetic analysis demonstrated that the IL-17Rs from other categories formed five major clades: IL-17RA, IL-17RB, IL-17RC, IL-17RD, and IL-17RE. Additionally, the SEFIR domain-contained proteins of crustaceans formed a unique clade, into which *Lvit-*IL-17R was classified.

### 3.2. Phylogenetic Analysis of IL-17 Domain-Contained Proteins from Decapods

After excluding identical amino acid sequences, we obtained 29 proteins from decapods containing the IL-17 domain from the NCBI database. Interestingly, most of these sequences are annotated as “uncharacterized protein”, rather than CFSH (Table A3). By comparing these sequences with the other CFSHs from previous studies [30], a limited number of sequences (GenBank accession nos. XP_042859036.1, XP_027234405.1, XP_027214658.1, XP_042860457.1, XP_042860456.1, XP_027214657.1, XP_027219797.1, XP_050718041.1, XP_042863675.1, and XP_027237919.1) could be annotated as CFSH. In addition, the IL-17 proteins (GenBank accession nos. XP_027214658.1, XP_027234405.1, XP_027214657.1, XP_027219797.1, and XP_027237919.1) from Pacific whiteleg shrimps, *P. vannamei,* have been identified as CFSH of the species in previous studies (Table A3) [30,31].

A phylogenetic tree was then constructed with the IL-17 domain-contained proteins of crustaceans (CFSH and other sequences) and the IL-17 from other categories (Figure 1B). Phylogenetic analysis demonstrated that the IL-17 domain-contained proteins of crustaceans were aggregated into a cluster, indicating that CFSH is an IL-17 homolog of decapod crustaceans.

### 3.3. Expression Profiles of Lvit-IL-17R

*Lvit-IL-17R* is widely expressed in various tissues (Figure 2A). *Lvit-IL-17R* in AG, hepatopancreas, and ovarian regions has different expression profiles.

In the present study, the *Lvit-IL-17R* expression level in the AG significantly decreased by 53.74% at stage II. Subsequently, it significantly increased to 143.39% of the stage I level during stage III. Finally, the *Lvit-IL-17R* level expression decreased to 78.68% of the stage I level during stage IV (*F*_3,15_ = 36.246, *p* < 0.05) (Figure 2B).

In the hepatopancreas, the *Lvit-IL-17R* expression level significantly increased to 314.84% at stage II, and then slightly reduced to 265.63% of the stage I level, and eventually, at stage IV, it downregulated to almost the same levels (111.76%) as those during stage I (*F*_3,15_ = 12.274, *p* < 0.05) (Figure 2B).

In the ovarian regions, the *Lvit-IL-17R* mRNA expression levels increased gradually with gonadal development. The *Lvit-IL-17R* mRNA expression levels significantly increased to 338.37% of the stage I level during stage II. The increase continued, with a peak expression at stage III at 916.07% of the stage I level. With the maturation of ovarian regions, at stage IV, the *Lvit-IL-17R* expression level significantly decreased to 413.87% of the stage I level (*F*_3,15_ = 57.831, *p* < 0.05) (Figure 2B).

### 3.4. Ligand–Receptor Interaction Analysis

The analysis of ligand–receptor interaction was performed through the utilization of GST pulldown assays. Through prokaryotic expression, recombinant proteins CFSH1a (21.0 kDa, containing 6 × His tag), CFSH1b (24.8 kDa, containing 6 × His tag), and IL-17R (62.8 kDa, containing 6 × His tag) (Figure 3) were obtained. The results of the GST pulldown assays demonstrate that rCFSH1a and rCFSH1b can bind to rIL-17R specifically rather than GST (Figure 3).

### 3.5. Short-Term Silencing Experiment In Vivo

The results demonstrate that compared with the PBS treatment, the transcripts levels of *Lvit-IL-17R* were inhibited by 56.2%, 46.6%, and 47.7% in the AG (*F*_2,12_ = 7.491, *p* < 0.05) (Figure 4A), hepatopancreas (*F*_2,12_ =20.662, *p* < 0.05) (Figure 4B), and ovarian regions (*F*_2,12_ = 22.197, *p* < 0.05) (Figure 4C), respectively.

The expression levels of the genes related to male sexual differentiation and ovarian development were also examined. Based on qRT-PCR, *Lvit-IL-17R* knockdown induced the significant upregulation of *Lvit-IAG1* (*F*_2,12_ = 64.934, *p* < 0.05) (Figure 4D) and *Lvit-IAG2* levels (*F*_2,12_ = 56.374, *p* < 0.05) (Figure 4E) in the AG, indicating that *Lvit-IL-17R* may suppress male sexual differentiation by inhibiting *IAG* expression. Following the knockdown of *Lvit-IL-17R*, the expression of *Lvit-Vg* (*F*_2,12_ = 20.633, *p* < 0.05) (Figure 4F) and *Lvit-VgR* (*F*_2,12_ = 43.263, *p* < 0.05) (Figure 4G) were also downregulated, suggesting the potential role of *Lvit*-IL-17R in ovarian development.

### 3.6. Long-Term Silencing Experiment In Vivo

Firstly, we evaluated the efficacy of gene knockdown in a long-term experiment. The injection of dsRNA successfully induced the knockdown of *Lvit-IL-17R* in the AG, hepatopancreas, and ovarian regions. Compared with the PBS treatment, the *Lvit-IL-17R* transcripts were inhibited by 77.7%, 73.6%, and 73.8% in the AG (*F*_2,14_ = 31.259, *p* < 0.05) (Figure 5A), hepatopancreas (*F*_2,14_ = 57.096, *p* < 0.05) (Figure 5B), and ovarian regions (*F*_2,14_ = 16.343, *p* < 0.05) (Figure 5C), respectively (Figure 5). In addition, it was observed that the silencing of *Lvit-IL-17R* had no discernible impact on the growth of the shrimps (carapace length: *F*_2,14_ = 0.172, *p* = 0.844 > 0.05; body weight: *F*_2,14_ = 1.057, *p* = 0.374 > 0.05) (Figure 5D,E).

#### 3.6.1. Effects of *Lvit-IL-17R* Silencing on Female Sexual Differentiation

The retardation of female sexual traits (female gonopores) was observed in the absence of *Lvit-IL-17R*. In the PBS and dsRNA *GFP* treatments, female gonopores were two distinct bulges located on the coxa of the third pair of pereiopods. The top edge of the female gonopores was surrounded by lush plumose setae. Following long-term *Lvit-IL-17R* silencing, the female gonopores became flattened and visually inapparent. The feathery setae surrounding female gonopores became sparse and were relatively short (Figure 6A).

The results demonstrated that *Lvit-IL-17R* silencing impeded the process of ovarian development. In the dsRNA *Lvit-IL-17R* treatment, the ovarian regions became smaller and filled with less-developed germ cells (Figure 6B). Relatively smaller oocytes and fewer follicular cells were observed during the dsRNA *Lvit-IL-17R* treatment (Figure 6B). We then measured the average oocyte diameter of the three treatments. The average oocyte diameters in both control groups were 58.89 ± 0.47 μm (n = 5) and 59.05 ± 1.06 μm (n = 6). This metric decreased to 37.00 ± 0.95 μm after *Lvit-IL-17R* silencing (n = 6) (*F*_2,14_ = 201.459, *p* < 0.05) (Figure 6C). Meanwhile, subsequent to the knockdown of *Lvit-IL-17R*, the genes associated with ovarian development were also significantly repressed. The *Lvit-Vg* (*F*_2,14_ = 244.880, *p* < 0.05) and *Lvit-VgR* (*F*_2,14_ = 17.844, *p* < 0.05) expression levels were significantly downregulated by 99.72% and 65.83%, respectively (Figure 6D).

#### 3.6.2. Effects of *Lvit-IL-17R* Silencing on Male Sexual Differentiation

Per the short-term silencing experiment, *Lvit-*IL-17R knockdown resulted in a notable elevation in the expression levels of *Lvit-IAG1* (*F*_2,14_ =8.667, *p* < 0.05) and *Lvit-IAG2* (*F*_2,14_ =33.143, *p* < 0.05) (Figure 7A).

Moreover, *Lvit-IL-17R* knockdown partially contributed to the development of male external phenotypes. *Lvit-IL-17R* gene knockdown significantly promotes appendices masculinae (AM) development (*F*_2,14_ = 6.676, *p* < 0.05) (Figure 7B,C). However, there were no significant variations observed in the other external male phenotypes, such as the male gonopores and cincinnuli (Figure 7B).

Meanwhile, *Lvit-IL-17R* knockdown dramatically promotes testicular development. Gonadal histology analysis revealed that the injection of *Lvit-IL-17R* dsRNA yielded different germ cell compositions. For both control groups, the majority of the cells were poorly developed germ cells, including spermatogonia (Sg) and spermatocytes I (Sc I), indicating less active spermatogenesis (Figure 7D). On the contrary, a more mature profile of germ cell compositions was established with the dsRNA *Lvit-IL-17R* treatment. Abundant mature germ cells, such as the spermatid (Sd) and spermatozoa (Sz), were observed in the testicular regions (Figure 7D).

## 4. Discussion

In both dioecious and hermaphrodite crustaceans, IAG and CFSH play pivotal roles in sexual differentiation [9,10,11,22,23]. IAG receptors have been identified in several crustacean species [18,19,20,21]. On the other hand, research regarding CFSH receptors remains scarce. In 2023, a candidate CFSH receptor for regulating IAG expression was identified from mud crabs, *S. paramamosain* [37]. We are unsure whether the receptors with similar structures regulate sexual differentiation in shrimps. The present study identified a CFSH receptor from PSH shrimps, *L. vittata*.

Previous studies have shown that the IL-17 domain is highly conserved in the mature peptides of all known CFSHs [24,25,26,27,28,29,30]. Also, we found that the recently identified IL-17 proteins were indeed CFSHs described in previous studies (Table A3) [30,31]. Thus, we suspected that CFSH was the IL-17 homolog in decapod crustaceans. To further confirm this suspicion, we constructed a phylogenetic tree with the IL-17 domain-contained proteins of crustaceans (CFSH and other sequences) and IL-17 from other categories (Figure 1B). Via phylogenetic analysis, we demonstrated that the IL-17 domain-contained proteins of crustaceans form a unique clade similar to other known IL-17. Thus, it is reasonable to suggest that CFSH is the IL-17 in decapod crustaceans.

Various IL-17Rs have been identified in vertebrates, from IL-17RA to IL-17RE [33,39,40]. In addition, the N-terminal regions of IL-17Rs are also varied in invertebrates [31,41,42]. IL-17 binds to different IL-17 receptors by triggering downstream signals [33,39,40,43]. Moreover, IL-17 receptors are all single-pass transmembrane receptors with a conversed SEFIR domain in the intracellular regions [31,33,36,39,40,41,42]. Therefore, the transcript of the SEFIR domain-contained protein was successfully obtained for further analysis. RACE cloning and bioinformatics analyses confirmed that the SEFIR-contained protein shares a similar structure with the other IL-17 receptors, including a signal peptide, a transmembrane domain, and an SEFIR domain at the cytoplasmic tail [33,36,39,40]. Thus, the transcript was named *Lvit-IL-17R*, rather than NF-κB activator 1 (Act1), a cytosolic protein containing the SEFIR domain [33]. Moreover, a recently identified CFSH receptor from mud crabs (*S. paramamosain)* also has a similar structure to those of the other IL-17R [37]. Thus, *Lvit-IL-17R* was speculated to be a candidate CFSH receptor of the PSH species.

By examining the tissue expression profile, *Lvit-IL-17R* was confirmed to be widely distributed in various tissues. Previous studies have also demonstrated that CFSH could suppress IAG expression in the AG and stimulate *Vg* expression in the hepatopancreas and *VgR* expression in the ovarian region, indicating that the AG, the hepatopancreas, and the ovarian region were target tissues of CFSH in these particular species [10]. To elucidate this, we examined the expression profiles in these three tissues. The results demonstrated that *Lvit-IL-17R* exhibited similar temporal expression profiles of *Lvit-CFSH1b* in the hepatopancreas and the ovarian region. However, *Lvit-IL-17R* displayed the opposite expression trend to those of *Lvit-IAG1* and *Lvit-IAG2*, which are more highly expressed in the male phase rather than the female phase [11]. The present findings suggest that *Lvit-IL-17R* may play a role in CFSH’s inhibition of IAG and CFSH’s promotion of ovarian development.

To explore the interaction between *Lvit*-IL-17R and *Lvit-*CFSHs, we performed GST pulldown assays with rCFSH1s and the extracellular segment of IL-17R. The results revealed that the extracellular segment of *Lvit*-IL-17R could bind to both *Lvit-*CFSHs. These findings suggest that *Lvit*-IL-17R is likely a receptor of type I CFSH in PSH shrimps. 

According to the previous studies, both type I CFSHs (CFSH1a and CFSH1b) co-regulate sexual differentiation in *L.vittata* [10]. In detail, CFSH1a and CFSH1b regulate the development of female gonopores. Furthermore, CFSH1b has been proposed to regulate ovarian development via vitellogenesis [10]. Although CFSH1a and CFSH1b suppress *IAG* expression in the AG, only CFSH1b is closely related to male sexual differentiation in the species [10]. Thus, an in vivo silencing experiment was conducted to explore whether *Lvit*-IL-17R had any functional relevance with CFSHs in the PSH species.

In the short-term silencing experiment, during the subsequent knockdown of *Lvit-IL-17R*, there was a notable increase in the expression levels of both *Lvit-IAGs*, whereas *Lvit-Vg* and *Lvit-VgR* were downregulated. Similar expression trends were observed for these genes in a previous study involving *Lvit-*CFSH1b [10]. These results suggest a correlation between *Lvit-*CFSH1b and *Lvit*-IL-17R in ovarian development and male sexual differentiation in this crustacean species.

To clarify the regulation roles of *Lvit*-IL-17R in sexual differentiation, we performed a long-term silencing experiment utilizing shrimps at the early gonadal development stages. The results indicate that *Lvit-IL-17R* knockdown hindered the development of female phenotypes (female gonopores), suggesting its regulatory role in female sexual differentiation. Moreover, the development of ovarian regions was also suppressed. Following *Lvit-IL-17R* knockdown, the ovarian regions were less developed with significantly smaller oocytes. Simultaneously, the transcripts of genes related to ovarian development were also inhibited. Similar results have been produced from the knockdown of either *CFSH1a* or *CFSH1b* in previous studies [10]. These findings suggest a relationship between *Lvit-*IL-17R and the two type I CFSHs (CFSH1a and CFSH1b) aimed at the female sexual differentiation of the PSH species.

Previous studies have shown that *Lvit*-IAG1 and *Lvit*-IAG2 co-regulate the development of male external phenotypes and spermatogenesis in PSH shrimps, *L. vittata* [11]. As IAG is regulated by type I CFSH through a negative feedback loop in both dioecious and hermaphrodite crustaceans [10,23], male sexual differentiation is also negatively regulated by CFSH1b in *L. vittata* [10]. The present results demonstrate that *Lvit-IL-17R* negatively regulates male sexual differentiation in this species. *Lvit-IL-17R* knockdown induces the promotion of the development of both testicular regions and male-related external phenotypes. These results also correspond to studies involving CFSH1b [10]. The combined results suggest that *Lvit*-IL-17R is a receptor of CFSH1b in the sexual differentiation of the PSH species.

Moreover, *Lvit*-IL-17R is also related to the biological functions of CFSH1a. A previous study demonstrated that *Lvit*-CFSH1a could also regulate female and male sexual differentiation by inhibiting *Lvit-IAG2* expression. Following *Lvit-CFSH1a* knockdown, the development of female gonopores was hindered, and the expression of *Lvit-IAG2* was promoted in the AG [10]. In the present study, *Lvit-IL-17R* knockdown also resulted in similar results. Moreover, *Lvit*-IL-17R could also bind to *Lvit*-CFSH1a. These findings suggest that *Lvit-IL-17R* is also a receptor of *Lvit-*CFSH1a.

## 5. Conclusions

In summary, a receptor of type I CFSH aimed at regulating sexual differentiation was identified through protein interaction and biological function experiments. To the best of the authors’ knowledge, the present study provides the first report on CFSH receptors involved in sexual differentiation in PSH species. We confirmed that type I CFSHs promote the development of both female external features and ovarian regions via *Lvit*-IL-17R in the PSH species. Moreover, *Lvit*-IL-17R is involved in the inhibition of *IAG* through CFSH, suppressing male sexual differentiation. These findings expand our understanding of crustacean reproductive endocrinology and clarify the mechanisms of sexual differentiation in crustaceans.

## Figures and Tables

**Figure 1 biomolecules-13-01456-f001:**
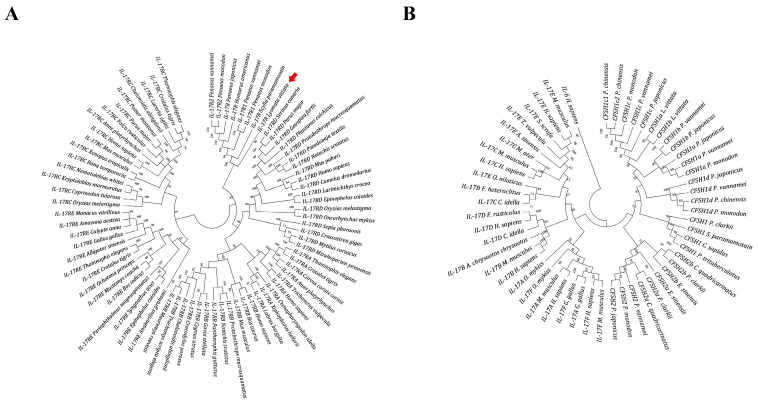
Phylogenetic tree of IL-17Rs (**A**) and CFSHs (**B**). Phylogenetic analysis was conducted using the Neighbor-Joining method, which is based on the Poisson model of MEGA7. The sequences used in the phylogenetic tree analysis of IL-17Rs and CFSHs are listed in Table A2 and Table A3, respectively. The numbers indicate bootstrap values based on 1000 replicates, as shown next to the branches. *Lvit-*IL-17R is indicated by a red solid arrow.

**Figure 2 biomolecules-13-01456-f002:**
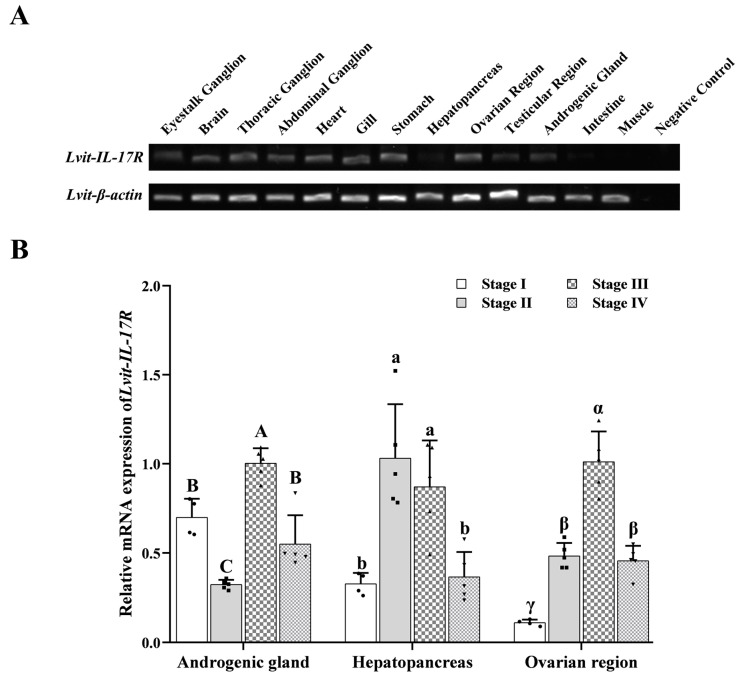
Spatial and temporal expression profiles of *Lvit-IL-17R* in *L. vittata*. (**A**) The tissue distribution profile of *Lvit-IL-17R* was generated using the RT-PCR assays of shrimps at the gonadal development of stage II. *Lvit-β-actin* was used as a positive control. (**B**) The expression profiles of *Lvit-IL-17R* in the androgenic gland, hepatopancreas, and ovarian regions during gonadal development through qRT-PCR. The *Lvit-IL-17R* expression levels were standardized using *Lvit-β-actin* expression levels (“A, B and C”, “a and b”, an “α, β, and γ”; *p* < 0.05; n = 4–5). Original WB images can be found in Appendix A.

**Figure 3 biomolecules-13-01456-f003:**
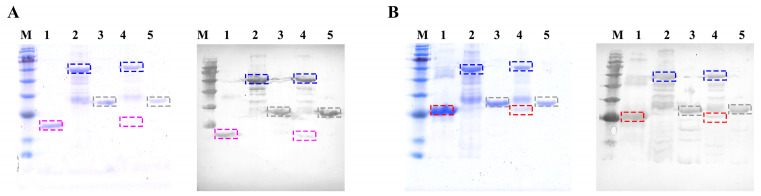
Pulldown assays conducted to detect the interaction between rCFSHs and *Lvit-IL-17R*. (**A**) Pulldown assays with rCFSH1a and rLvit-IL-17R. Lane M; protein marker; Lane 1: initially purified rCFSH1a; Lane 2: initially purified r*Lvit*-IL-17R; Lane 3: initially purified GST protein; Lane 4: Eluent of beads after co-incubation with rCFSH1a and r*Lvit*-IL-17R; Lane 5: Eluent of beads after co-incubation with GST and r*Lvit*-IL-17R. (**B**) Pulldown assays with rCFSH1b and r*Lvit*-IL-17R. Lane M; protein marker; Lane 1: initially purified rCFSH1b; Lane 2: initially purified r*Lvit*-IL-17R; Lane 3: initially purified GST protein; Lane 4: Eluent of beads after co-incubation with rCFSH1b and r*Lvit*-IL-17R; Lane 5: Eluent of beads after co-incubation with GST and r*Lvit*-IL-17R. rCFSH1a, ~21.0 kDa, marked with purple dashed box; rCFSH1b, ~24.8 kDa, marked with red dashed box; GST protein, ~28.3 kDa, marked with gray dashed box; rLvit-IL-17R, ~62.8 kDa, marked with blue dashed box. Original WB images can be found in Appendix A.

**Figure 4 biomolecules-13-01456-f004:**
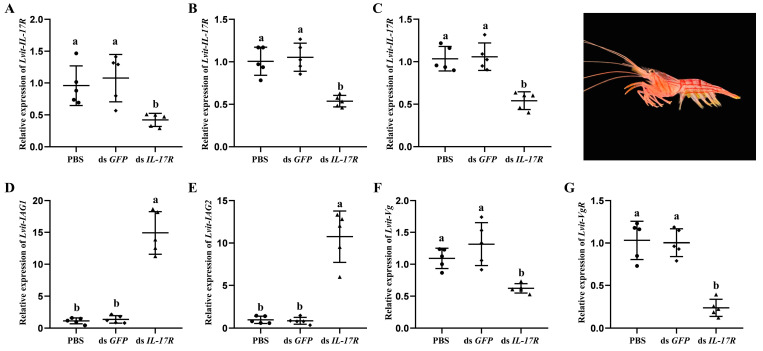
Short-term silencing experiment in vivo. The effectiveness of gene knockdown was evaluated in the androgenic gland (**A**), hepatopancreas (**B**), and ovarian regions (**C**). The expression of genes related to sexual differentiation (*Lvit-IAG1* and *Lvit-IAG2*) (**D**,**E**) and ovarian development (*Lvit-Vg* and *Lvit-VgR*) (**F**,**G**) were also detected. The gene expression levels were standardized by *Lvit-β-actin* expression levels and represented as mean ± SD (“a and b”, *p* < 0.05; n = 5).

**Figure 5 biomolecules-13-01456-f005:**
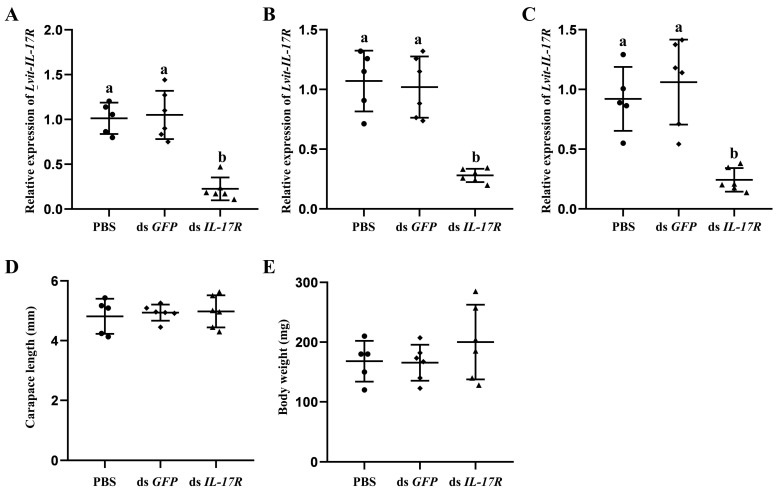
Long-term silencing experiment in vivo. The effectiveness of gene knockdown was evaluated in the androgenic gland (**A**), hepatopancreas (**B**), and ovarian regions (**C**). The effect of *Lvit-IL-17R* knockdown on *L. vittata* growth was also assessed. The carapace length (**D**) and body weight (**E**) were measured (“a and b”, *p* < 0.05; n = 5–6).

**Figure 6 biomolecules-13-01456-f006:**
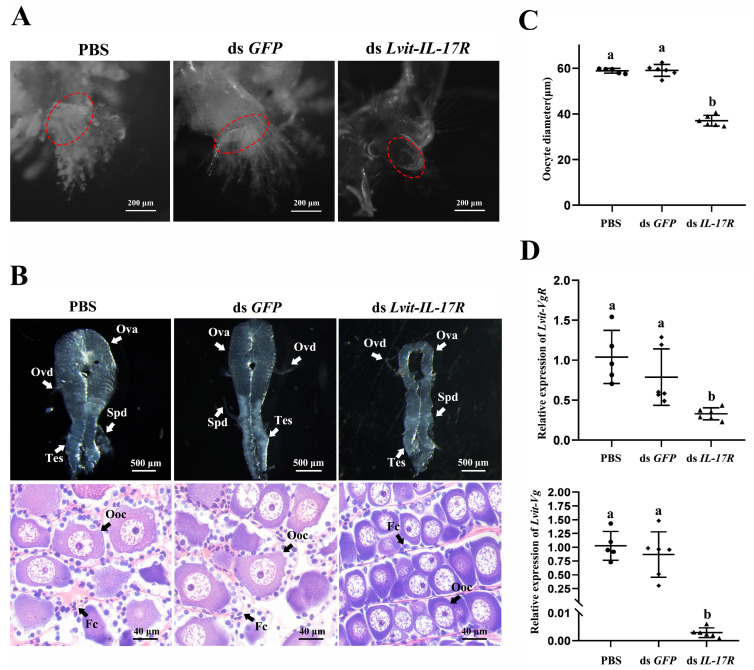
Effect of long-term *Lvit-IL-17R* silencing on female sexual differentiation. (**A**) Effect of long-term *Lvit-IL-17R* silencing on female gonopores. Female gonopores were marked with red dotted circles. (**B**) Effect of long-term *Lvit-IL-17R* silencing on ovarian development. Ovotestes were photographed to examine tissue morphology. The histological features were further analyzed via H & E staining. Ovd: oviduct; Ova: ovary; Spd: sperm duct; Tes: testis; Ooc: oocytes; Fc: follicular cell. (**C**) The long and short axis lengths of each oocyte were measured and averaged, yielding the diameter for each cell (“a and b”, *p* < 0.05; n = 5–6). (**D**) Expression levels of *Lvit-VgR* in ovarian regions and *Lvit-Vg* in hepatopancreas were also detected (“a and b”, *p* < 0.05; n = 5–6).

**Figure 7 biomolecules-13-01456-f007:**
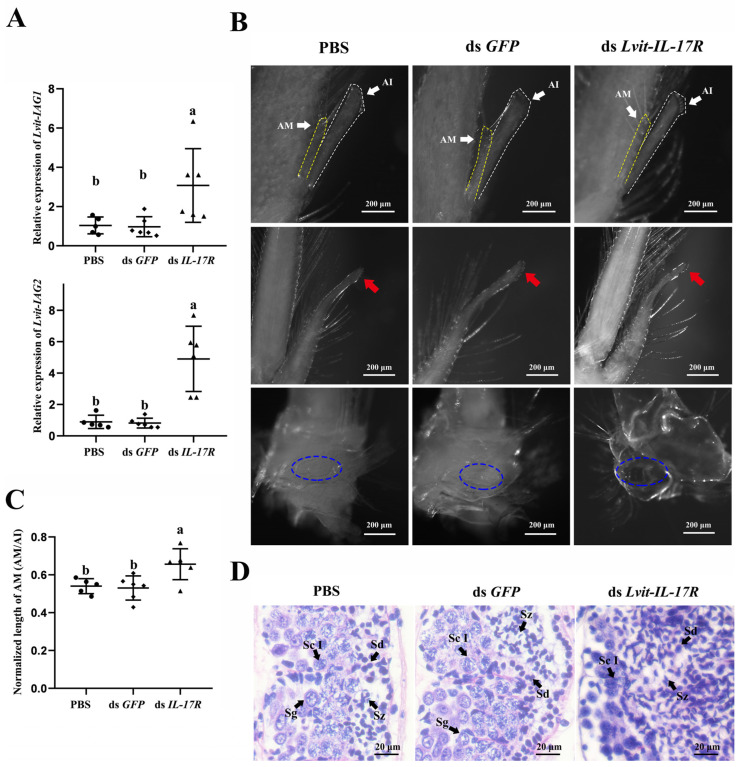
Effect of long-term *Lvit-IL-17R* silencing on male sexual differentiation. (**A**) Expression levels of *Lvit-IAG1* and *Lvit-IAG2* in androgenic gland were detected (“a and b”, *p* < 0.05; n = 5–6). (**B**) Effect of long-term *Lvit-IL-17R* silencing on the development of male external phenotypes. AM and AI are represented by yellow and white dash lines, respectively. Cincinnuli are marked with red solid arrows. Male gonopores were indicated by blue dotted circles. AM: appendices masculinae; AI: appendix interna. (**C**) The length of AM and AI were measured, and the normalized length of AM (AM/AI) was calculated (“a and b”, *p* < 0.05; n = 5–6). (**D**) Effect of long-term *Lvit-IL-17R* silencing on testicular development. H & E staining was employed to analyze histological features of testicular regions. Sg: spermatogonia; Sc I: primary spermatocyte; Sd: spermatid; Sz: spermatozoa.

## Data Availability

The original contributions presented in the study are publicly available. This data can be found here: (GenBank accession no. MZ367742).

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
