# Peer review of "Identification of Crustacean Female Sex Hormone Receptor Involved in Sexual Differentiation of a Hermaphroditic Shrimp"

_biomolecules, 2023, doi:10.3390/biom13101456_

Round 1
Author Response
Response to Reviewer 1 Comments
Thank you for your constructive comments. We are appreciated to have a chance to improve our manuscript. Below we address your comments in a point-to-point response.
Point 1: First, from a nomenclature viewpoint, it is confusing that while the crab and shrimp receptors are structurally homologous to each other and to the class of IL-17 receptors, they chose the name Sp- SEFIR for the crab one and Lvit-IL-17R for the shrimp one. It would be good to unify the nomenclature (I think IL-17-R is clearer, they may want to suggest renaming the crab one).
Response 1: Thanks for your suggestions. We will rename the crab one “Sp-IL-17R” and reload sequence to the Genebank.
Point 2: Line 12: “hermaphroditism” should be “hermaphroditic”.
Response 2: We have changed "hermaphroditism" to "hermaphroditic". Please see line 12.
Point 3: Line 31: “and it can be classified into two types”: it is not clear what “it” refers to.
Response 3: We are sorry for the unclear expression. “it” refers to the intersex species. Changes have been made accordingly. Please see line 30-31.
Point 4: 3.1 Sequence analysis of Lvit-IL-17-R
The authors should show a schematic view of the domain structure of the receptor, similar to Fig. 1A in the IJMS 2003 paper (ref. 41). Also maybe give the % homology of each subdomain between the crab and the shrimp.
I would also like to see the cDNA sequence and deduced amino acid sequence like in appendix B of ref. 41.
Response 4: Thanks for your suggestions. cDNA sequence and deduced amino acid sequence of Lvit-IL-17R, % homology of crab and shrimp, and schematic diagram of Lvit-IL-17R were provided. Please see Figure A1A, A1B and line 196-198, 494-501.
Point 5: -3.3 Expression profiles of Lvit-IL-17R
Lines 223-231: the description of the expression levels at the 4 stages of development in Figure 2 is clearly inadequate. Regarding AG, rather than “decreased sharply” at stage II, give the level of stage II expression in % of stage I, like in the description of figures 4 and 5. Rather than “dramatically increased to top levels” at stage III, give % of stage I level (it is clearly not “top”, it is less than stage I. “…later decreased to same levels as stage I at stage IV”: this is clearly wrong, it is even less than at stage III: again give exact %. Regarding the hepatopancreas, “levels increased dramatically to top levels at stage II” is obviously wrong, it looks like at most 23% and may not be significantly different given the large SD. “later decreased gradually” is also wrong, stage IV is equal or even slightly higher than stage III. For ovarian region, also give % figures rather than “gradually” and “sharp”. The statistical significance of the differences between the bars is not shown. 2
Response 5: We are deeply sorry for the mistake we made. Temporal expression profiles of Lvit-IL-17R were examined for several times. As we upload a mismatched figure, Figure 2B does not match the text description. We have replaced Figure 2B to the one that matches data analysis and the text description. We have rephrased this section accordingly. Please see Figure 2B, and line 231-244.
Point 6: In Fig. 2, the meaning of the letters (capitals, lower case or greek alphabet) on top of the bars is not described anywhere.
Expressions like F3,15 = 36.246 here and elsewhere in the text are not defined and their meaning is not clear.
Why do the authors use colored bar graphs and show the experimental points in Fig 4, 5 and 6 and not in Fig. 2?
Response 6: We are sorry for missing content of statistical analyses. F value is one factor analysis of variance (One – way ANOVA). We have added corresponding contents. Please see line 185-190.
As we upload a mismatched figure, Figure 2B does not show the experimental points. We have have replaced Figure 2B to the one that show the experimental points. Please see Figure 2B.
Point 7: 3.5 Short-term silencing experiment in vivo
Lines 273-274: “… might male sexual differentiation…” should be “… might induce male sexual differentiation…”.
Response 7: Thanks for your suggestions. We have changed it to "may suppress male sexual differentiation". Please see line 322.

Reviewer 2 Report
The authors show the role of IL-17 signaling in the sex determination of L. vittata. It is already reported that CFSH, a neurohormone with a highly coserved IL-17 domain, has a crucial role both in blocking spermatogenesis and male sexual phenotype in hermaphroditic shrimp. In this study, the authors identified the sequence of the homolog gene of Lvit-IL-17R in L. vittata, and showed that its downregulation induced some abnormality in the ovarian development. It is interesting result, but there are some
1. The author showed that the IL-17R silencing in L. vittata induced IAG downregulation and irregular ovarian phenotype, such as smaller oocyte, and fewer follicular cells, but did not show the effect of CFSH. Certainly, CFSH has an IL-17 domain and binds to IL-17R, but the author did not show that CFSH is the only factor binding to IL-17R in the gonad of L.vittata. Therefore, the title and conclusion regarding the role of CFSH are inadequate. If the author claims the role of CFSH, the knockdown analysis of CFSH should also be shown.
2. The crucial roles of CFSH and IL-17R in the other shrimp species are already reported. What is the novelty or difference from the other shrimp in gonadal development? Or, The author should describe more how important or meaningful it is to identify the role of CFSH in the female sexual development of PSH.
3. The effect of IL-17R silencing on female external phenotype is very important data in this study. The author should describe the phenotype in more detail. The description in lines 300-303 and Figure 7 is too simple. For example, where are the luch feathery setae and bulged out like a frustum in Figure 7?
The editing of the English language by a native should be required.
Author Response
Response to Reviewer 2 Comments
Thank you for your constructive comments. We are appreciated to have a chance to improve our manuscript. Below we address your comments in a point-to-point response.
Point 1: The author showed that the IL-17R silencing in L. vittata induced IAG downregulation and irregular ovarian phenotype, such as smaller oocyte, and fewer follicular cells, but did not show the effect of CFSH. Certainly, CFSH has an IL-17 domain and binds to IL-17R, but the author did not show that CFSH is the only factor binding to IL-17R in the gonad of L.vittata. Therefore, the title and conclusion regarding the role of CFSH are inadequate. If the author claims the role of CFSH, the knockdown analysis of CFSH should also be shown.
Response 1: Thanks for your suggestions. We have changed tittle to "Identification of Crustacean Female Sex Hormone Receptor Involved in Sexual Differentiation of a Hermaphroditic Shrimp". Please see line 2-3.
Point 2: The crucial roles of CFSH and IL-17R in the other shrimp species are already reported. What is the novelty or difference from the other shrimp in gonadal development? Or, The author should describe more how important or meaningful it is to identify the role of CFSH in the female sexual development of PSH.
Response 2: To date, the crucial roles of CFSH in sexual differentiation have been demonstrated in a few species, including the dioecious species (the Atlantic blue crab C. sapidus, the mud crab S. paramamosain and the Chinese mitten crab Eriocheir sinensis) and the hermaphroditic species (the peppermint shrimp L. vittata).
It is not conclusive yet that what role CFSH plays in the process of ovarian development. In the Atlantic blue crab C. sapidus, CFSH was suggested not related to ovarian development, whereas CFSH was suggested to regulate some unknown physiological process other than vitellogenesis in the kuruma prawn M. japonicus and the giant freshwater prawn M. rosenbergii. Instead, our previous studies indicated that CFSH could directly stimulate ovarian development via promoting vitellogenesis in the hermaphroditic shrimp L. vittata. Sequences of IL-17R in the other shrimps are all from genome and transcriptome studies, and their crucial roles have never been studied yet.
In the PSH shrimp L. vittata, individuals are all intersexual with ovotestis and androgenic gland (the sex-determination organ in the dioecious species) throughout their life. Gonadal development, including development of ovarian regions and testicular regions, is co-regulated by both male and female regulatory factors.
Point 3: The effect of IL-17R silencing on female external phenotype is very important data in this study. The author should describe the phenotype in more detail. The description in lines 300-303 and Figure 7 is too simple. For example, where are the luch feathery setae and bulged out like a frustum in Figure 7?
Response 3: Thanks for your suggestions. According to previous studies, CFSH's roles in development of female phenotypes were mainly focus on gonopores and ovigerous setae (he Atlantic blue crab C. sapidus, the mud crab S. paramamosain and the Chinese mitten crab Eriocheir sinensis). Our previous studies indicated that CFSHs regulate development of female gonopores. It is not clear whether development of ovigerous setae was mainly regulated by CFSH of the PSH species. Thus, we only choose female gonopores as the female external phenotype in long-term silencing experiment. Herein, we have rephrased this section accordingly. Please see line 309-315.
Point 4: The editing of the English language by a native should be required.
Response 4: Thanks for your suggestions. We have carefully edited the entire manuscript and the manuscript has been polished by a professional editor before resubmission.

Round 2
Reviewer 1 Report
The authors have made a major revision of their paper, not only by accommodating my recommendations for improvement but also by carefully revising the writing and language, and reorganizing the figures. They also made the title more explicit. I think the paper quality has been much improved to a high level.
Reviewer 2 Report
I feel the paper is ready for publication
I will entrust the editor with everything related to English language.